



# A Systematic Approach to Offshore Wind Turbine Jacket Pre-Design and Optimization: Geometry, Cost, and Surrogate Structural Code Check Models

Jan Häfele[1], Rick Damiani[2], Ryan King[2], Cristian G. Gebhardt[1], and Raimund Rolfes[1]

[1]Leibniz Universität Hannover, Institute of Structural Analysis, Appelstr. 9a, 30167 Hannover, Germany
[2]National Renewable Energy Laboratory, 15013 Denver West Parkway, Golden, CO 80401, United States of America
*Correspondence to:* Jan Häfele (j.haefele@isd.uni-hannover.de)

**Abstract.** The main obstacles in preliminary design studies or optimization of jacket substructures for offshore wind turbines are high numerical expenses for structural code checks and simplistic cost assumptions. In order to create a basis for fast design evaluations, this work provides the following: first, a jacket model is proposed that covers topology and tube sizing with a limited set of design variables. Second, a cost model is proposed that goes beyond the simple and common mass-dependent approach. And third, the issue of numerical efficiency is addressed by surrogate models both for fatigue and ultimate limit state code checks. In addition, this work shows an example utilizing all models. The outcome can be utilized for preliminary design studies and jacket optimization schemes and is suitable for scientific and industrial applications.

**Nomenclature**

| | |
|---|---|
| DLC | Design load case |
| $\mathbb{E}$ | Expected value |
| $\mathcal{GP}(m,k)$ | Gaussian process with mean function $m$ and covariance function $k$ |
| GPR | Gaussian process regression |
| MSL | Mean sea level |
| $\mathbb{N}$ | Set of natural numbers |
| $\mathcal{N}(\mu,\sigma^2)$ | Normally distributed number with mean $\mu$ and variance $\sigma^2$ |
| SF | Partial safety factor |
| TI | Turbulence intensity |
| $\Phi_p$ | Planar (two-dimensional) batter angle |
| $\Phi_s$ | Spatial (three-dimensional) batter angle |
| $\beta_b$ | Brace-to-leg diameter ratio at bottom (jacket model parameter) |
| $\beta_i$ | Brace-to-leg diameter ratio in the $i$th bay |
| $\beta_t$ | Brace-to-leg diameter ratio at top (jacket model parameter) |
| $\gamma_b$ | Leg radius-to-thickness ratio at bottom (jacket model parameter) |
| $\gamma_i$ | Leg radius-to-thickness ratio in the $i$th bay |





| | | |
|---|---|---|
| | $\gamma_t$ | Leg radius-to-thickness ratio at top (jacket model parameter) |
| | $\theta_{wave}$ | Wave direction |
| | $\theta_{wind}$ | Wind direction |
| | $\xi$ | Head-to-foot radius ratio (jacket model parameter) |
| 5 | $\rho$ | Material density (jacket model parameter) |
| | $\sigma_n^2$ | Gaussian input noise variance |
| | $\vartheta$ | Angle enclosed by two jacket legs |
| | $\tau_b$ | Brace-to-leg thickness ratio at bottom (jacket model parameter) |
| | $\tau_i$ | Brace-to-leg thickness ratio in the $i$th bay |
| 10 | $\tau_t$ | Brace-to-leg thickness ratio at top (jacket model parameter) |
| | $\psi_{1,i}$ | Lower brace-to-leg connection angle in the $i$th bay |
| | $\psi_{2,i}$ | Upper brace-to-leg connection angle in the $i$th bay |
| | $\psi_{3,i}$ | Brace-to-brace connection angle in the $i$th bay |
| | $C_j$ | Expenses related to $j$th cost factor |
| 15 | $C_{total}$ | Total capital expenses |
| | $D_{Bb}$ | Bottom brace diameter |
| | $D_{Bt}$ | Top brace diameter |
| | $D_L$ | Leg diameter (jacket model parameter) |
| | $E$ | Material Young's modulus (jacket model parameter) |
| 20 | $G$ | Material shear modulus (jacket model parameter) |
| | $H_s$ | Significant wave height |
| | $\mathbf{I}$ | Identity martrix |
| | $\mathbf{K}$ | Kernel function matrix |
| | $L$ | Overall jacket length (jacket model parameter) |
| 25 | $L_{MSL}$ | Transition piece elevation over MSL (jacket model parameter) |
| | $L_{OSG}$ | Lowest leg segment height (jacket model parameter) |
| | $L_{TP}$ | Transition piece segment height (jacket model parameter) |
| | $L_i$ | $i$th jacket bay height |
| | $L_{m,i}$ | Distance between the lower layer of K-joints and the layer of X-joints of the $i$th bay |
| 30 | $M$ | Size of the cross-validation leftover |
| | $N$ | Number of cross-validation bins |
| | $N_L$ | Number of legs (jacket model parameter) |
| | $N_X$ | Number of bays (jacket model parameter) |
| | $P$ | Probability density function |
| 35 | $R_{Foot}$ | Foot radius (jacket model parameter) |
| | $R_{Head}$ | Head radius |
| | $R_i$ | $i$th jacket bay radius at lower K-joint layer |





| | | |
|---|---|---|
| | $R_{m,i}$ | Radius of the $i$th X-joint layer |
| | $T_{Bb}$ | Bottom brace thickness |
| | $T_{Bt}$ | Top brace thickness |
| | $T_{Lb}$ | Bottom leg thickness |
| 5 | $T_{Lt}$ | Top leg thickness |
| | $T_p$ | Wave peak period |
| | $\mathbf{X}$ | Matrix of training inputs (one sample per row) |
| | $a$ | Kernel weighting parameter |
| | $a_j$ | $j$th unit cost |
| 10 | $c_j$ | $j$th cost factor |
| | $e$ | Noise |
| | $e_{bias}$ | Bias error |
| | $e_{mse}$ | Mean square error |
| | $f$ | Function value |
| 15 | $\boldsymbol{k}$ | Kernel function vector |
| | $k$ | Covariance (kernel) function |
| | $k_{Ma_{3/2}}$ | Matérn 3/2 kernel function |
| | $k_{Ma_{5/2}}$ | Matérn 5/2 kernel function |
| | $k_{RQ}$ | Rational quadratic kernel function |
| 20 | $k_{SE}$ | Squared exponential kernel function |
| | $l$ | Kernel length-scale parameter |
| | $m$ | Mean function |
| | $q$ | Ratio of two consecutive bay heights (jacket model parameter) |
| | $u_{ss}$ | Sub-surface current velocity |
| 25 | $u_w$ | Near-surface current velocity |
| | $v_s$ | Mean wind speed |
| | $\boldsymbol{x}$ | Array of design variables/Vector of training inputs |
| | $\boldsymbol{x}^*$ | Array of prediction inputs |
| | $x_{MB}$ | Mud brace flag (jacket model parameter) |
| 30 | $\boldsymbol{y}$ | Vector of training outputs (one sample per row) |
| | $y$ | General regression output value |
| | $y^*$ | Prediction value |





# 1 Introduction

In the oil and gas industry, the jacket substructure is well-established due to a good trade-off between cost efficiency and reliability. It has been considered for offshore wind turbine substructures for several years and has already had some successful applications in Europe and the United States. Smith et al. (2015) showed that among all wind farms announced to be built from the second quarter of 2015 until 2020, $16\%$ of the substructures are jackets, whereas this share was only $10\%$ for wind farms built before 2015 (see Figure 1 for the market shares of offshore wind turbine substructures in the past and nowadays). Despite potential advantages, the market is still strongly dominated by monopiles (Ho et al., 2016), as financial aspects and significantly lower uncertainty play an important role from an economical point of view (BVGassociates, 2012). However, the development of new turbines with higher rated power in combination with the need for deeper water installations might be a catalyst for a technological leap toward jacket substructures. Damiani et al. (2016) calculated that for water depths deeper than $40$-m jackets promise lower costs than monopiles, considering six offshore sites along the U.S. Eastern Seaboard and the Gulf of Mexico. The break-even point or water depth, respectively, where the jacket technology becomes truly competitive, is however dependent on the costs of the vessels used to transport and install the structures. State-of-the-art jackets can still benefit from design studies and structural optimization to render lower costs to the project (BVGassociates, 2013), which is addressed by current research. The accumulation of publications dealing with this topic in the recent past is an indication of this statement. Chew et al. (2016) and Oest et al. (2016) performed structural optimization of jacket substructures with simulation-based approaches using gradient-based algorithms. Basis for these papers was the structure defined in the first phase of the Offshore Code Comparison Collaboration Continuation (OC4) project (Popko et al., 2014). Damiani et al. (2017) studied the impact of environmental and turbine parameters on the costs or mass, respectively, of jackets, considering $81$ different structures. Hübler et al. (2017) analysed the effect of variations in jacket design on the economic viability. AlHamaydeh et al. (2017) and Kaveh and Sabeti (2018) used meta-heuristic algorithms for the optimization of jacket substructures, however, without realistic—in particular, fatigue limit state—load assumptions. Stolpe and Sandal (2018) introduced discrete variables in the jacket optimization problem formulation to account for the fact that steel tubes are only available in fixed dimensions.

From a global perspective, the main obstacles that lead to nonoptimal structures are both the dependence on expert knowledge and the large computational cost associated with the optimization of a complex structure. Many design assessments or optimization approaches addressing this problem fail (because they lead to either unrealistic or impractical design) for the following simple reasons:

– **Design variables**. Most approaches do not consider the structural topology, but only the sizing of predefined members. Involving topological parameters, which may be real or discrete, as design variables is mandatory for a proper design that makes use of a mixed-integer formulation.

– **Cost assumptions**. Often, the mass of the entire jacket is used as an objective function in optimization approaches. But obviously, the cost breakdown for a welded structure includes many items which do not depend on the mass of the structure. Moreover, other expenses such as transport and installation costs, should not be ignored.



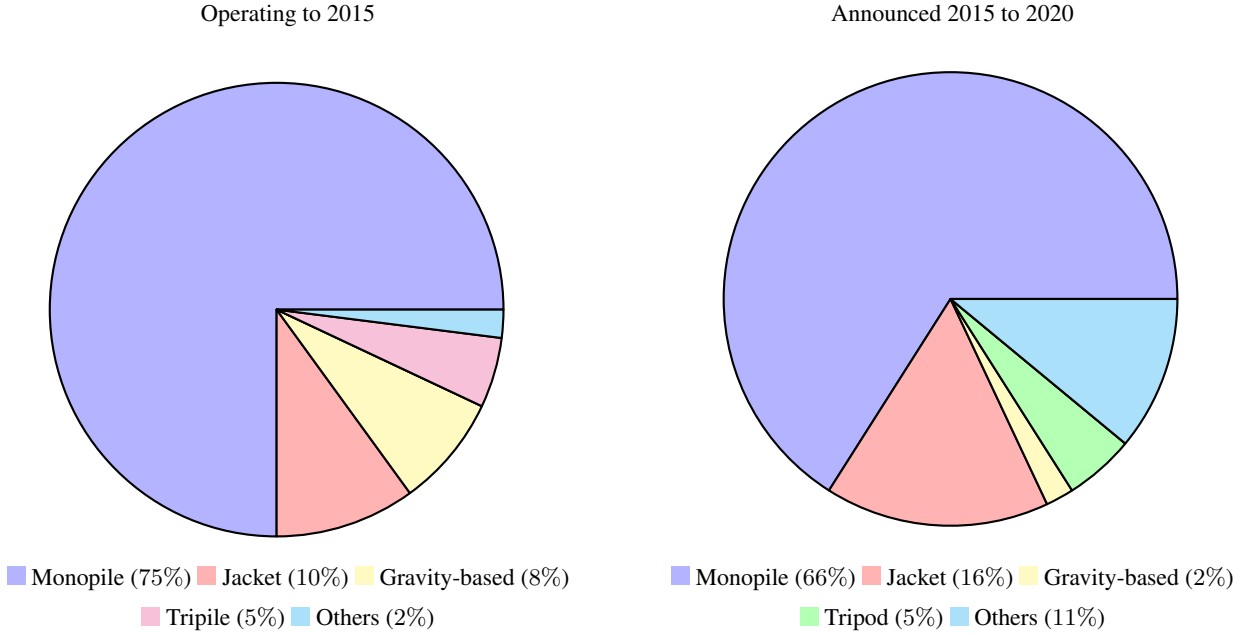

**Figure 1.** Share of utilized substructures among operating turbines in 2015 and from 2015 to 2020 according to Smith et al. (2015).

- **Load assumptions**. The assumption of simplified environmental states (for instance, the omission of wind-wave mis-alignment) is the state of the art in many jacket design procedures, because it relaxes the computational demand and fills any existing gap in the knowledge of the actual metocean conditions.

- **Structural code checks**. A realistic jacket design involves structural design code checks for fatigue and ultimate limit
state based on time domain simulations. Many approaches miss either one or both of them, most likely because the computational implementation is resource intensive.

- **Simulation approaches**. Design iterations cause changes in the structural behavior. A coupled simulation or at least a rigorous approach addressing this aspect is mandatory. However, it is often seen that sequential approaches are applied where decoupled loads are exchanged at the interface between substructure and turbine tower, even in the case of fatigue
assessment.

One possible approach to address some of these issues was the jacket sizing tool proposed by Damiani and Song (2013), which enables the conceptual design considering preliminary load assumptions. It, however, lacked extensions to full dynamics simulations and fatigue limit states. Wind turbine cost models are available (Fingersh et al., 2006; National Wind Technology Center Information Portal, 2014) and were used for the definition of wind turbine optimization objectives and constraints
(Ning et al., 2013), but without explicit or detailed cost formulations for jacket substructures. The goal of the current work is to provide a basic jacket model that can be efficiently used in conceptual studies and optimization approaches by preventing the issues stated above and providing a basis for more realistic designs and mainly using mathematically manageable equations.





The first part of this study addresses the first two points described above. The last three points are handled in the second part, as they involve a completely different field. This paper is structured as follows: Section 2 explains the utilized jacket model with the assumptions made for the structural details. In Section 3, a simple cost model is proposed, which covers cost contributions from materials, fabrication, transition piece, coating, and transport and installation (including foundation). In Section 4, load

sets are defined for both fatigue and ultimate limit state load cases and a design of experiments is created to fit appropriate surrogate models. The paper concludes with remarks on the benefits of the jacket model, its limitations, and a brief outlook on further work based on this model.

## 2 Jacket Model

The previous section summarized some issues leading to certain requirements of a simple jacket model:

– The set of design variables must be as comprehensive as necessary to accurately model the fundamental topology, physics, and dynamics of a typical jacket but as small as possible for ease of computation, too.

– The design variables must cover both topological and geometrical parameters.

– Structural details with little bearing on the mechanical behavior shall be disregarded.

– The cost model formulation shall only depend on the parameters of the jacket model.

– The structure shall be manufacturable, transportable, and installable.

– The structure shall be easily transferable to common design tools (mostly based on finite element formulations).

A concept matching all of these points was initially described by Häfele and Rolfes (2016) and is extended in this section. At first, the topology is defined; then the tube dimensions and material properties are derived.

### 2.1 Topology

The main presumption is that the jacket model need not be limited to a certain number of legs or brace layers (bays), but instead allows different topologies. As foot and head girth are measures related to four-legged structures, a general formulation in terms of foot (on the ground layer) and head (on the same layer as the transition piece) circles with foot and head radii, $R_{Foot}$ and $R_{Head}$, respectively, is introduced. In order to prevent obtaining structures with a funnel shape, a parameter, $\xi$, is introduced, which relates both radii (and can be set to a value less than or equal to 1). The two circles depict the bottom and

top of a frustum of length, $L$ (see Figure 2, first step). The $N_L$ legs can then be constructed as straight lines on the surface of the cone, equidistantly distributed. This step is illustrated for a four-legged jacket in the second step of Figure 2. However, this procedure is applicable to every number of legs that is greater than or equal to three. With these variables, the angle enclosed by two legs can be found according to the following equation:

$$\vartheta = \frac{2\pi}{N_L}.$$ (1)





The spatial batter angle, $\Phi_s$, is the inclination angle of each leg with respect to the symmetry axis of the frustum (sometimes denoted as the three-dimensional batter angle):

$$\Phi_s = \arctan\left(\frac{R_{foot}\,(1-\xi)}{L}\right). \tag{2}$$

The planar batter angle, $\Phi_p$, is the inclination angle projected to a vertical-horizontal layer through the symmetry axis of the frustum (sometimes denoted as the two-dimensional batter angle):

$$\Phi_p = \arctan\left(\frac{R_{foot}\,(1-\xi)\sin\left(\frac{\vartheta}{2}\right)}{L}\right). \tag{3}$$

The parameter, $N_X$, defines the number of bays. A bay is one part of the jacket that is delimited by $N_L$ single- or double-K-joints at the lower side and $N_L$ single- or double-K-joints at the upper side and comprises all structural elements in between, in particular $N_X$ X-joints. The $i$th bay is denoted with $i$, where:

$$i \in \mathbb{N}^{[1,N_X]}. \tag{4}$$

The ratio, $q$, relates the heights of two consecutive bays, $L_{i+1}$ and $L_i$, which is assumed to be constant:

$$q = \frac{L_{i+1}}{L_i}. \tag{5}$$

It has to be noted that $L_1$ is the height of the lowest bay and $L_{N_X}$ is the height of the highest one. Based on the previous assumptions and elementary geometrical considerations, circles on every K-joint layer can be constructed. With the height of the entire jacket, $L$, the distance between the ground and lowest bay, $L_{OSG}$, and the distance between the transition piece and highest bay, $L_{TP}$, the $i$th jacket bay height, $L_i$, can be calculated by:

$$L_i = \frac{L - L_{OSG} - L_{TP}}{\sum_{n=1}^{N_X} q^{n-i}}. \tag{6}$$

The radius of each bay (at the lower K-joint layer), $R_i$, is:

$$R_i = R_{foot} - \tan\left(\Phi_s\right)\left(L_{OSG} + \sum_{n=1}^{i-1} L_n\right). \tag{7}$$

This third step is shown in Figure 2. The distance between the lower layer of K-joints and the layer of X-joints for the $i$th bay, $L_{m,i}$, can be calculated by simple geometrical relations:

$$L_{m,i} = \frac{L_i R_i}{R_i + R_{i+1}}. \tag{8}$$

The radius of the $i$th X-joint layer is:

$$R_{m,i} = R_{foot} - \tan\left(\Phi_s\right)\left(L_{OSG} + \sum_{n=1}^{i-1} L_n + L_{m,i}\right). \tag{9}$$





The lower and upper brace-to-leg connection angles, $\psi_{1,i}$ and $\psi_{2,i}$, respectively, and the brace-to-brace connection angle, $\psi_{3,i}$, in the $i$th bay are related as follows:

$$\psi_{1,i} = \frac{\pi}{2} - \arctan\left(\frac{R_{foot}\,(1-\xi)\sin\left(\frac{\vartheta}{2}\right)\cos\left(\Phi_p\right)}{L}\right) - \arctan\left(\frac{L_{m,i}}{R_i\sin\left(\frac{\vartheta}{2}\right)\cos\left(\Phi_p\right)}\right), \tag{10}$$

$$\psi_{2,i} = \frac{\pi}{2} + \arctan\left(\frac{R_{foot}\,(1-\xi)\sin\left(\frac{\vartheta}{2}\right)\cos\left(\Phi_p\right)}{L}\right) - \arctan\left(\frac{L_{m,i}}{R_i\sin\left(\frac{\vartheta}{2}\right)\cos\left(\Phi_p\right)}\right), \tag{11}$$

$$\psi_{3,i} = 2\arctan\left(\frac{L_{m,i}}{R_i\sin\left(\frac{\vartheta}{2}\right)\cos\left(\Phi_p\right)}\right). \tag{12}$$

In addition, $L_{MSL}$ is the distance between the transition piece (which is on the same height as the tower foot) and mean sea level layer or—in other words—the difference between jacket length and water depth. This information is necessary to create a mesh for the computation of hydrodynamic loads. The flag, $x_{MB}$, determines whether the jacket is equipped with mud braces or not. The final topology (step four) is illustrated in Figure 2, in this example with four legs ($N_L = 4$), four bays ($N_X = 4$), and a mud brace ($x_{MB} = \text{true}$).

## 2.2 Tube Dimensions

The proposed jacket model makes no use of prefabricated joints (as in the state of the art), therefore no joint cans or stiffeners (mainly to improve punching shear resistance) are used. The consequences of only single-sided welds and no stiffened joints should be considered. However, the number of (expensive) welds is reduced to a minimum, which reduces the number of degrees of freedom in a structural analysis as well. Moreover, the cost model is not burdened by possible impacts of series production for prefabricated joints. This is not far away from practical application: it was analyzed for substructures with a rated power higher than $10\,\mathrm{MW}$ in the research project INNWIND.EU and evaluated as the most efficient one concerning fabrication costs (Scholle et al., 2015).

Instead of regarding the diameters and thicknesses of each tube as independent variables, which would lead—depending on the structural topology—to a high number of design variables, the tube dimensions are interpolated between values at the top and the bottom of the structure. Another potential problem is that the tube dimensions, if all are regarded as independent, might lead to undesirable relations between the tube dimensions. Standards and guidelines provided by DNV GL AS (2016); **?** for the design and certification of offshore structures propose the adoption of three ratio parameters, initially defined by Efthymiou (1988). However, one variable has to be independent; in our case, the leg diameter, $D_L$, which is assumed to be constant.

$\gamma_b$ and $\gamma_t$ define the ratios between leg radii (not the diameters) and thicknesses:

$$\gamma_b = \frac{D_L}{2T_{Lb}}, \tag{13}$$

$$\gamma_t = \frac{D_L}{2T_{Lt}}, \tag{14}$$



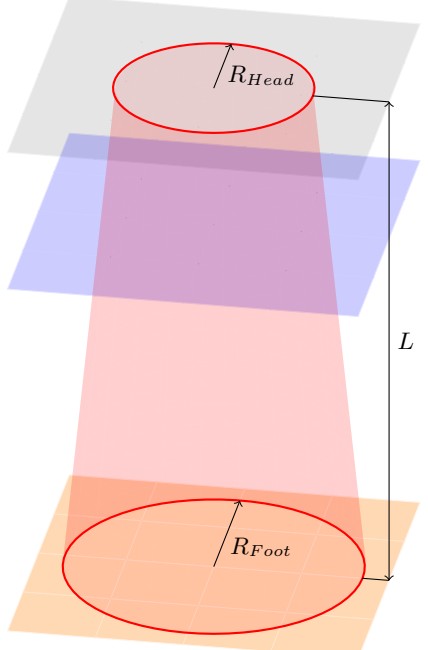

(1) Truncated cone defined by $R_{Foot}$, $R_{Head}$, and $L$

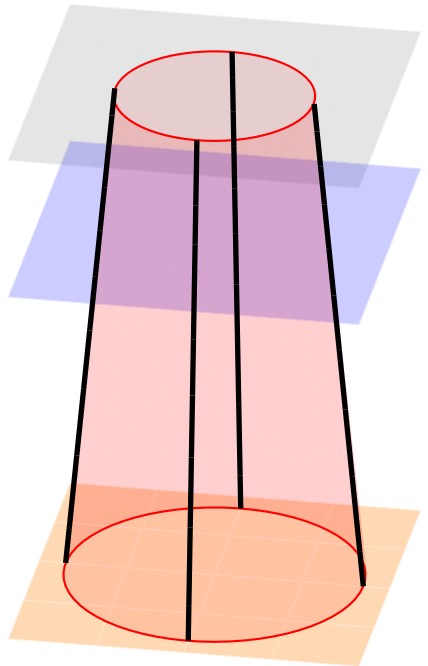

(2) Creating $N_L$ jacket legs, here: $N_L = 4$

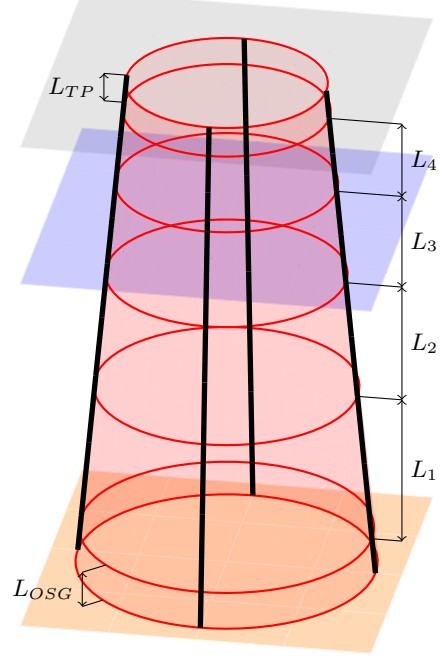

(3) Creating $N_X + 1$ K-joint layers, here: $N_X = 4$

(4) Final jacket with braces

**Figure 2.** Creation of the jacket topology in four steps.



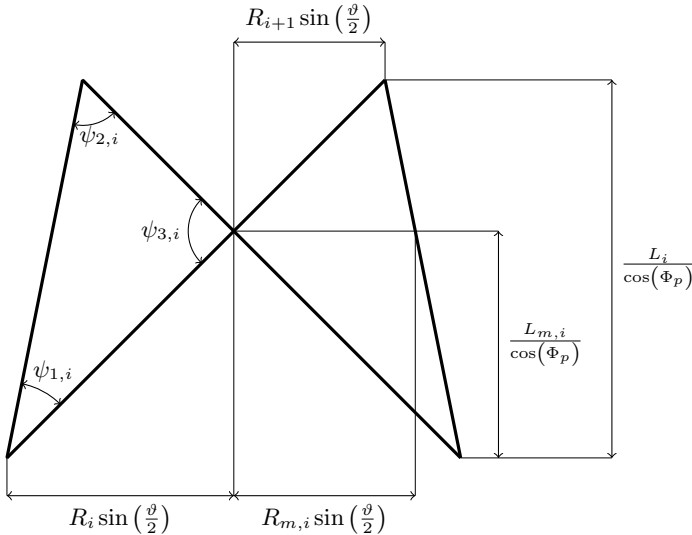

**Figure 3.** $i$th jacket bay topology projected to the layer of X- and K-joints on one side of the structure.

where the index, $b$, indicates the affiliation to the lowermost (bottom) and $t$ to the uppermost (top) tubes. The parameters, $\beta_b$ and $\beta_t$, define the ratios of brace and leg diameter at the bottom and top, respectively:

$$\beta_b = \frac{D_{Bb}}{D_L}, \tag{15}$$

$$\beta_t = \frac{D_{Bt}}{D_L}. \tag{16}$$

5   The values $\tau_b$ and $\tau_t$ define the relations between brace and leg thicknesses at the bottom and top, respectively:

$$\tau_b = \frac{T_{Bb}}{T_{Lb}}, \tag{17}$$

$$\tau_t = \frac{T_{Bt}}{T_{Lt}}. \tag{18}$$

The final determination of the leg and brace dimensions as functions of the height elevation is illustrated in Figure 4 and 5. As no tapered tubes are desirable, a linear-stepwise interpolation is performed, therefore:

$$\gamma_i = \begin{cases} \gamma_b & i = 1 \\ \frac{\gamma_t - \gamma_b}{L - L_{N_X} + L_{m,N_X} - L_{TP}} \left( L_{OSG} + \sum_{n=1}^{i-1} L_n + L_{m,i} \right) + \gamma_b & \text{else,} \end{cases} \tag{19}$$

$$\beta_i = \frac{\beta_t - \beta_b}{L - L_{N_X} - L_{OSG} - L_{TP}} \sum_{n=1}^{i-1} L_n + \beta_b, \tag{20}$$

$$\tau_i = \frac{\tau_t - \tau_b}{L - L_{N_X} - L_{OSG} - L_{TP}} \sum_{n=1}^{i-1} L_n + \tau_b. \tag{21}$$



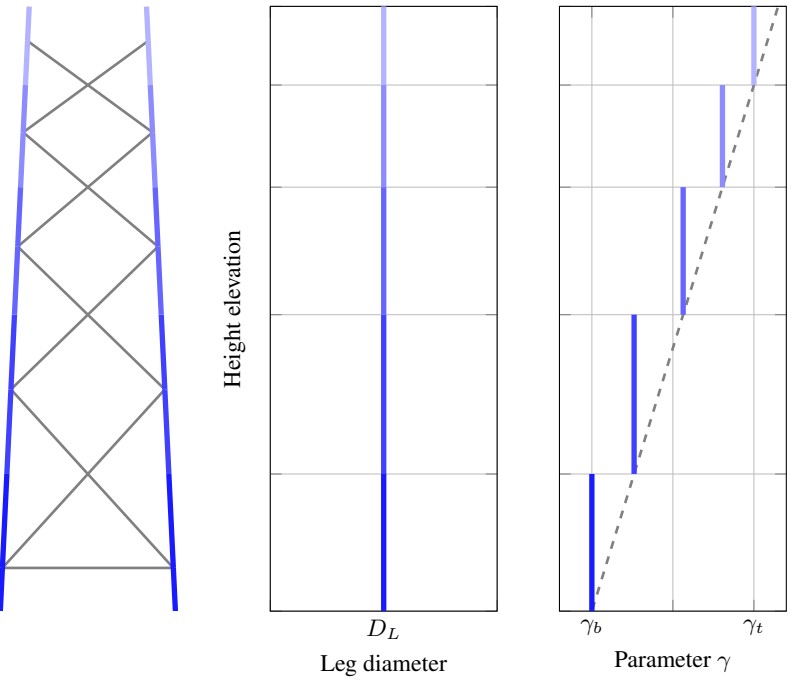

**Figure 4.** Definition of leg dimensions in dependency of the jacket height. Values are illustrated by ▬ at the bottom and shade to ▬ at the top of the structure.

## 2.3 Material Properties

The entire jacket is supposed to be made of the same isotropic material, which can be described by the Young's modulus, $E$, the shear modulus, $G$, and the material density, $\rho$.

## 2.4 Parameter Summary and Array of Design Variables

5   There are 20 parameters of the jacket model in total, where ten describe the topology, seven the tube dimensions, and three the material properties. It can be assumed that site- and material-dependent parameters are commonly predetermined, so the number of free design variables might be smaller than 20. To ease the notation in what follows, all variables of the jacket model are assembled in the array, $\boldsymbol{x}$:

$$\boldsymbol{x} = (N_L \ N_X \ R_{foot} \ \xi \ L \ L_{MSL} \ L_{OSG} \ L_{TP} \ x_{MB} \ q \ D_L \ \gamma_b \ \gamma_t \ \beta_b \ \beta_t \ \tau_b \ \tau_t \ E \ G \ \rho)^T . \tag{22}$$

10   **3   Cost Modeling**

A possible approach to the jacket substructure cost calculation is to regard the total capital expenses, $C_{total}$, as a linear combination of multiple contributions, where each one is given by a cost factor, $c_j$, multiplied by the corresponding unit cost $a_j$:



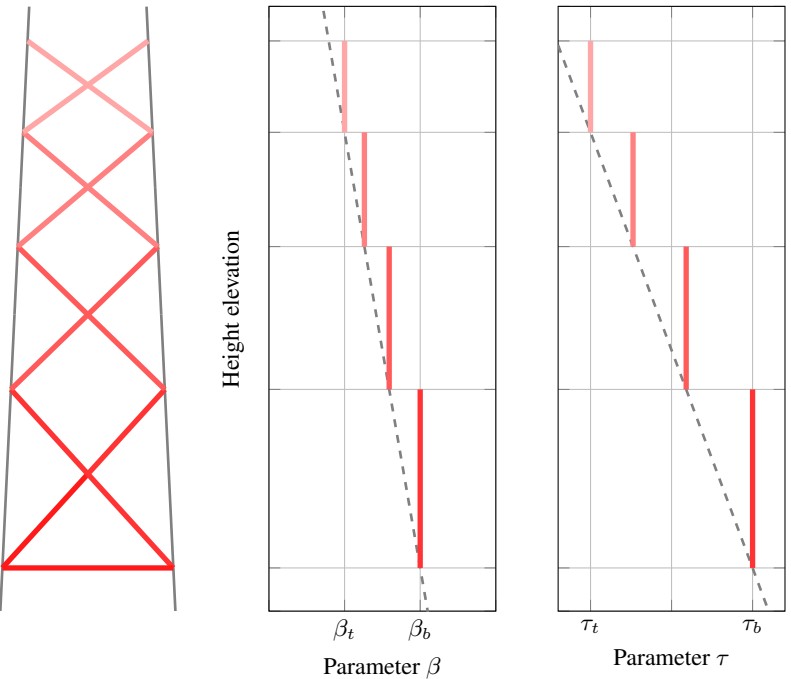

**Figure 5.** Definition of brace dimensions in dependency of the jacket height. Values are illustrated by ▬▬ at the bottom and shade to ▬▬ at the top of the structure.

$$C_{total}(\boldsymbol{x}) = \sum \underbrace{a_j c_j(\boldsymbol{x})}_{C_j(\boldsymbol{x})}. \tag{23}$$

Basic factors for material, fabrication, coating, transition piece, and structural appurtenances (if regarded as additional parts), and transport and installation (including costs for the pile foundation) are assumed here, acknowledging that this breakdown may look different if increasing the level of detail:

$c_1$:     Material factor

$c_2$:     Fabrication factor

$c_3$:     Coating factor

$c_4$:     Transition piece factor

$c_5$:     Transport factor

$c_6$:     Foundation and installation factor

$c_7$:     Fixed expenses factor.





## 3.1 Material Expenses

The material expenses are supposed to be proportional to the mass of the components. Therefore, $c_1$ is the total jacket mass which is the mass of the assembled substructure excluding the transition piece, foundation, or appurtenances of any kind and which can be obtained by evaluating structural analysis tools or by applying simple geometrical relations:

$$
\begin{aligned}
c_1(\boldsymbol{x}) =\, & 2\rho N_L \pi D_L^2 \sum_{i=1}^{N_X} \left( \left( \frac{\beta_i \tau_i}{2\gamma_i} + \frac{\tau_i^2}{4\gamma_i^2} \right) \sqrt{\frac{L_i^2}{\cos^2(\Phi_p)} + (R_i + R_{i+1})^2 \sin^2\left(\frac{\vartheta}{2}\right)} \right) \\
& + x_{MB}\rho N_L \pi D_L^2 \left( \frac{\beta_b \tau_b}{2\gamma_b} + \frac{\tau_b^2}{4\gamma_b^2} \right) 2R_1 \sin\left(\frac{\vartheta}{2}\right) \\
& + \rho N_L \pi D_L^2 \sum_{i=1}^{N_X} \left( \left( \frac{1}{2\gamma_i} + \frac{1}{4\gamma_i^2} \right) \frac{L_{m,i}}{\cos(\Phi_s)} + \left( \frac{1}{2\gamma_{i+1}} + \frac{1}{4\gamma_{i+1}^2} \right) \frac{(L_i - L_{m,i})}{\cos(\Phi_s)} \right) \\
& + \rho N_L \pi D_L^2 \left( \frac{1}{2\gamma_b} + \frac{1}{4\gamma_b^2} \right) \frac{L_{OSG}}{\cos(\Phi_s)} \\
& + \rho N_L \pi D_L^2 \left( \frac{1}{2\gamma_t} + \frac{1}{4\gamma_t^2} \right) \frac{L_{TP}}{\cos(\Phi_s)}.
\end{aligned}
\tag{24}
$$

## 3.2 Fabrication Expenses

Although it can be assumed that fabrication expenses contribute significantly to the overall jacket costs, this factor is often neglected, because it is difficult to measure. A common approach in practical applications is to assume a proportional relation to the cumulated weld volume. In this cost model, $c_2$ is the cumulative volume of all structural welds. With the weld root thickness, $t_0$ (given as $3\,\mathrm{mm}$ in Germanischer Lloyd (2012)), and assuming a $45°$ weld angle around the entire weld, the sectional weld area can be approximated and multiplied by the weld length, which is the perimeter of the ellipse that is projected to the connected chord surface[1], thus:

$$
\begin{aligned}
c_2(\boldsymbol{x}) =\, & 2 N_L \pi D_L \sum_{i=1}^{N_X} \left( \beta_i \left( \frac{D_L^2 \tau_i^2}{8\gamma_i^2} + \frac{t_0 D_L \tau_i}{2\sqrt{2}\gamma_i} \right) \left( \sqrt{\frac{1}{2\sin^2(\psi_{1,i})} + \frac{1}{2}} + \sqrt{\frac{1}{2\sin^2(\psi_{2,i})} + \frac{1}{2}} + \sqrt{\frac{1}{2\sin^2(\psi_{3,i})} + \frac{1}{2}} \right) \right) \\
& + 2 x_{MB} N_L \pi D_L \beta_b \left( \frac{D_L^2 \tau_b^2}{8\gamma_b^2} + \frac{t_0 D_L \tau_b}{2\sqrt{2}\gamma_b} \right) \\
& + N_L \pi D_L \sum_{i=1}^{N_X} \left( \frac{D_L^2 \min\left(\frac{1}{\gamma_i^2}, \frac{1}{\gamma_{i+1}^2}\right)}{8} + \frac{D_L t_0 \min\left(\frac{1}{\gamma_i}, \frac{1}{\gamma_{i+1}}\right)}{2\sqrt{2}} \right).
\end{aligned}
\tag{25}
$$

The equation uses the perimeter of the ellipse that is projected on a plane to calculate the weld length. This is not exactly equal to the real weld length, but simplifies the equation considerably.

---

[1]The ellipse perimeter is approximated in a very simple way here. However, the occurring eccentricities are in a range where this simplification causes no significant error.



### 3.3 Coating Expenses

Coating is necessary to protect the jacket from corrosion and causes non-negligible costs. It is assumed that the entire outer surface area of all tubes is coated after production and the coating expenses are proportional to the outer surface area $c_3$:

$$c_3(\boldsymbol{x}) = N_L \pi D_L \left( 2 \sum_{i=1}^{N_X} \left( \beta_i \sqrt{\frac{L_i^2}{\cos^2(\Phi_p)} + (R_i + R_{i+1})^2 \sin^2\left(\frac{\vartheta}{2}\right)} \right) + x_{MB} \beta_b \left( 2R_1 \sin\left(\frac{\vartheta}{2}\right) \right) + \frac{L}{\cos(\Phi_s)} \right). \qquad (26)$$

### 3.4 Transition Piece Expenses

Although there are different transition piece types, a stellar-type transition piece is assumed, which connects the uppermost leg ends with straight bars to a center point. In this case, it can be assumed that the costs depend linearly on the one hand on the number of legs and on the other hand on the head radius, thus the factor $c_4$ reads:

$$c_4(\boldsymbol{x}) = N_L R_{foot} \xi. \qquad (27)$$

### 3.5 Transport Expenses

For a simplified cost estimation, the expenses to be raised for the transport of the structure from the port to the wind farm site can be roughly measured in terms of a linear mass dependency, therefore factors $c_5$ and $c_1$ are equal:

$$c_5(\boldsymbol{x}) = c_1(\boldsymbol{x}). \qquad (28)$$

However, this value (mass after production) is supposed to be slightly different from the wrought mass that is used due to overlapping joints and material removal prior to welding. To simplify the cost calculation, it is assumed that both values are equal.

### 3.6 Foundation and Installation Expenses

The foundation is the structural part that provides an interface to the seabed. Both the production costs for the foundation structures—no matter of which type—and the on-site installation costs depend linearly on the number of legs in our approach. For the sake of simplicity, it is assumed that these costs do not cover costs due to modifications of the structural pile design. They are assembled in the foundation and installation expenses and the corresponding factor $c_6$ reads:

$$c_6(\boldsymbol{x}) = N_L. \qquad (29)$$

### 3.7 Fixed Expenses

There are costs that cannot be measured in terms of any parameters of the jacket model, that is:

$$c_7(\boldsymbol{x}) = 1. \qquad (30)$$

These kinds of costs—in the nomenclature of this work proportional to the factor $c_7$—arise for every structure and are indeed very important for a cost assessment, but have a rather minor impact on the design studies or optimization results, as there is



no contribution to differential operators. Examples are costs for structural appurtenances, like boat landings and ladders, or production facilities and infrastructure, like scaffolds or cranes.

## 4 Surrogate Models for Fatigue and Ultimate Limit State

A general presupposition made in this work is that realistic jacket design necessitates simulation-based proofs to ensure the
structural functionality in different limit states. While the proof of serviceability limit state is mostly simple in the case of relatively stiff lattice structures, where the tubular tower dominates the modal behavior of the entire turbine, the checks for fatigue and ultimate limit state are computationally expensive. There are indeed simulation-based optimization approaches in the literature, but all with very limited design load sets, and proposals trying to find efficient load sets or simplifications on load cases.

Recent work showed that Gaussian process regression (GPR) models are appropriate to predict numerically obtained fatigue damages for two test structures from environmental state inputs (Brandt et al., 2017). It is thus straightforward to transfer the same methodology to the prediction of fatigue damages or utilization ratios due to extreme loads for varying jacket designs in case the load sets are given. It is also imaginable to apply a classification approach to this type of problem, with the statements "structural code check successful" or "structural code check failed" as outputs. However, this would limit the imaginable
applications, so regression is applied. In the following, a brief introduction into Gaussian process regression is given. For the sake of simplicity, the output dimension of the problem is restricted to one, that is a single-output regression problem. The basis for GPR is the Bayesian regression problem:

$$y = f(\boldsymbol{x}) + e \tag{31}$$

with

$$e \sim \mathcal{N}\left(0, \sigma_n^2\right). \tag{32}$$

We want to make predictions, $y^*$, for an arbitrary set of (prediction) input variables, $\boldsymbol{x}^*$, based on information gathered from the training set, which is represented by the input matrix, $\mathbf{X}$, and the vector of corresponding output values, $\boldsymbol{y}$. The key assumption of Gaussian process regression is that a Gaussian distribution over $f(\boldsymbol{x})$ exists, thus:

$$f(\boldsymbol{x}) \sim \mathcal{GP}\left(m(\boldsymbol{x}), k(\boldsymbol{x}, \boldsymbol{x}')\right), \tag{33}$$

with

$$m(\boldsymbol{x}) = \mathbb{E}\left[f(\boldsymbol{x})\right] \tag{34}$$

and

$$k(\boldsymbol{x}, \boldsymbol{x}') = \mathrm{cov}\left[f(\boldsymbol{x}), f(\boldsymbol{x}')\right], \tag{35}$$





which is a Mercer kernel function. Due to the marginalization property of Gaussian processes, there is a joint distribution of training and prediction sets:

$$\begin{pmatrix} \boldsymbol{y} \\ y^* \end{pmatrix} \sim \left( \boldsymbol{0}, \begin{bmatrix} \mathbf{K}(\mathbf{X},\mathbf{X}) & \boldsymbol{k}(\mathbf{X},\boldsymbol{x}^*) \\ \boldsymbol{k}(\boldsymbol{x}^*,\mathbf{X}) & k(\boldsymbol{x}^*,\boldsymbol{x}^*) \end{bmatrix} \right). \tag{36}$$

In this equation, $\mathbf{K}$ and $\boldsymbol{k}$ were introduced to ease the notation and just represent matrices and vectors, where each element is

the corresponding value of $k$. The mean of the joint distribution was set to zero. From this equation, the conditional posterior distribution of $y^*$ can be obtained:

$$y^*|\boldsymbol{x}^* \sim \mathcal{N}\left( \boldsymbol{k}(\mathbf{X},\boldsymbol{x}^*)^T \left(\mathbf{K}(\mathbf{X},\mathbf{X})+\sigma_n^2\mathbf{I}\right)^{-1}\boldsymbol{y}, \; k(\boldsymbol{x}^*,\boldsymbol{x}^*)-\boldsymbol{k}(\boldsymbol{x}^*,\mathbf{X})\left(\mathbf{K}(\mathbf{X},\mathbf{X})+\sigma_n^2\mathbf{I}\right)^{-1}\boldsymbol{k}(\mathbf{X},\boldsymbol{x}^*) \right). \tag{37}$$

For further details, the interested reader is referred to Rasmussen and Williams (2008), which is the most comprehensive work in this field in the opinion of the authors. Due to the probabilistic nature of these models, the computation of prediction intervals

is possible. This is a substantial advantage, because realistic load sets are large and thus the size of the design of experiments is limited. In addition, when the uncertainty arising from design load set assumptions is known, it can be easily considered by an appropriate choice and parametrization of the kernel function.

The prediction of values from a GPR model requires the complete input and output training to set it up. In contrast to the proposed geometry and cost assumptions, the derivation of surrogate models for fatigue and ultimate limit state depends highly

on the reference turbine and the environmental conditions. The first one has been selected to be the National Renewable Energy Laboratory (NREL) 5-MW turbine, defined by Jonkman et al. (2009). The water depth at the fictive location is $50\,\text{m}$. In addition, the research platforms FINO3 (mainly) and FINO1 (for validation purposes) provide detailed, long-term measurements to derive the environmental conditions. Soil properties are adopted from the definition of the soil layers in the Offshore Code Comparison Collaboration (OC3) project (Jonkman and Musial, 2010). There are, however, some limitations in these assump-

tions that cannot be suppressed. The assumption of $50$-m water depth does not match the water depths at the FINO locations. Nevertheless, no other measurements of environmental states are available, and this assumption was also made in the design basis of the UpWind project (Fischer et al., 2010).

### 4.1 Training and Validation Data Sets

To obtain training data for surrogate modeling, 200 test jackets were sampled from the design space by a Latin Hyper Cube

Sampling with minimum correlation between all samples. The boundaries given in Table 1 were chosen conservatively, already excluding "too optimistic"[2] jacket designs. Although the number of samples seems to be low, it has to be considered that the number of time domain simulations depends linearly on the sample size. Moreover, equation 37 requires the inversion of $\mathbf{K}(\mathbf{X},\mathbf{X})$, which may lead to weak numerical performance of the prediction. Furthermore, an independent validation set with 40 samples from the entire design space was generated, which was created by another Latin Hyper Cube sampling. It has to

be noted that the purpose of this data set is just validation of the final parameterized models, it is not involved in the training phase and is not part of the cross-validation procedure.

---

[2]This statement means that the structural code checks allow wider ranges of the design parameters.



**Table 1.** Jacket model parameter boundaries for design of experiments. Topological, tube sizing, and material parameters are separated in groups; single values state that the corresponding value is held constant.

| Parameter | Description | Lower Boundary | Upper Boundary |
|---|---|---|---|
| $N_L$ | Number of legs | 3 | 4 |
| $N_X$ | Number of bays | 3 | 5 |
| $R_{foot}$ | Foot radius | $6.792\,\mathrm{m}$ | $12.735\,\mathrm{m}$ |
| $\xi$ | Head-to-foot radius ratio | 0.533 | 0.733 |
| $L$ | Overall jacket length | $70.0\,\mathrm{m}$ | |
| $L_{MSL}$ | Transition piece elevation over MSL | $20.0\,\mathrm{m}$ | |
| $L_{OSG}$ | Lowest leg segment height | $5.0\,\mathrm{m}$ | |
| $L_{TP}$ | Transition piece segment height | $4.0\,\mathrm{m}$ | |
| $q$ | Ratio of two consecutive bay heights | 0.640 | 1.200 |
| $x_{MB}$ | Mud brace flag | true | |
| $D_L$ | Leg diameter | $0.960\,\mathrm{m}$ | $1.440\,\mathrm{m}$ |
| $\gamma_b$ | Leg radius-to-thickness ratio (bottom) | 12.0 | 18.0 |
| $\gamma_t$ | Leg radius-to-thickness ratio (top) | 12.0 | 18.0 |
| $\beta_b$ | Brace-to-leg diameter ratio (bottom) | 0.533 | 0.800 |
| $\beta_t$ | Brace-to-leg diameter ratio (top) | 0.533 | 0.800 |
| $\tau_b$ | Brace-to-leg thickness ratio (bottom) | 0.350 | 0.650 |
| $\tau_t$ | Brace-to-leg thickness ratio (top) | 0.350 | 0.650 |
| $E$ | Material Young's modulus | $2.100 \times 10^{11}\,\mathrm{N\,m^{-2}}$ | |
| $G$ | Material shear modulus | $8.077 \times 10^{10}\,\mathrm{N\,m^{-2}}$ | |
| $\rho$ | Material density | $7.850 \times 10^{3}\,\mathrm{kg\,m^{-3}}$ | |

## 4.2 Design Load Sets

In order to conduct time domain simulations, load sets both for fatigue and ultimate limit state have to be defined. For the fatigue case, a broad knowledge about the required size of design load sets is already available, because it was analyzed previously in a comprehensive study (Häfele et al., 2017a, b), where both probabilistic and unidirectional load sets were investigated. However, as the GPR allows to propagate uncertainties, it is reasonable to utilize a probabilistic load set with 128 production load cases (design load case (DLC) 1.2 and 6.4 according to IEC-61400-3, see International Electrotechnical Commision, 2009)) for damage estimation (see Table 2), which is a finding of the previously mentioned study. In the extreme load case, the focus is rather on the consideration of multiple special events than on the reproduction of the long-term behavior. Table 3 features a summary of all design load cases that are to be calculated for every sample. There are ten extreme load cases that were identified to be potentially critical. DLC 1.3 and 1.6a are production load cases with extreme turbulence and severe sea





**Table 2.** Considered design load sets according to IEC-61400-3 (International Electrotechnical Commision, 2009) for the fatigue limit state (SF: partial safety factor, $v_s$: mean wind speed, $P$: probability density function, TI: turbulence intensity, $H_s$: significant wave height, $T_p$: wave peak period, $\theta_{wind}$: wind direction, $\theta_{wave}$: wave direction, $u_w$: near-surface current velocity, $u_{ss}$: sub-surface current velocity, MSL: mean sea level). Yaw error is normally distributed with $-8°$ mean value and $1°$ standard deviation.

| DLC | Quantity | Wind | Waves | Directionality | Current | Water Level |
|---|---|---|---|---|---|---|
| 1.2, 6.4 | | $v_s = P(v_s)$ | $H_S = P(H_s\|v_s)$ | $\theta_{wind} = P(\theta_{wind}\|v_s)$ | $u_w(0) = 0.42\,\mathrm{m\,s^{-1}}$ | |
| | 128 | | | | | MSL |
| SF $= 1.25$ | | $\mathrm{TI} = \mathrm{TI}(v_s)$ | $T_p = P(T_p\|H_s)$ | $\theta_{wave} = P(\theta_{wave}\|H_s,\theta_{wind})$ | $u_{ss}(0) = 0\,\mathrm{m\,s^{-1}}$ | |

state, respectively. DLC 2.3 is a design load case, where electrical grid loss occurs during the production state. DLC 6.1a and 6.2a are events with extreme mean wind speed, the first one with an extreme sea state and the second one with an extreme yaw error.

### 4.3 Time Domain Simulations

As the varying jacket design changes the structural behavior of the entire turbine, only fully coupled simulations were conducted for this study, as so-called sequential or uncoupled approaches are considered not sufficently accurate. All simulations are computed with FAST (National Wind Technology Center Information Portal, 2016) in the current version at the publication of this study and comprise 10-min time series[3] plus an additional 3 min time for transient decay. Soil-structure interaction is considered in a reduced representation of the substructure (see Häfele et al., 2016), where eight interior modes are the basis for the representation of the jacket with foundation. The operating-point-dependent soil behavior cannot be neglected in the extreme load case and is considered by an ad-hoc approach (Hübler et al., 2016).

### 4.4 Post-Processing of Time Domain Results

Fatigue is evaluated in terms of the maximum cumulative damage that occurs in the critical joint after summing up all hot spot damages. An S-N curve approach defined by the structural code DNV GL RP-0005 (**?**) is utilized for this purpose. Stress cycles are evaluated by a Rainflow counting algorithm and added up according to linear damage accumulation. The jacket model assumes one-sided welds. In this case, an S-N curve with an endurance stress limit of $52.63 \times 10^6\,\mathrm{N\,m^{-2}}$ at $10^7$ cycles and slopes of 3 and 5 before and after endurance limit, respectively, was selected.

Ultimate limit state proofs are performed according to the structural code NORSOK N-004 (NORSOK, 2004), which is a well-established standard for this purpose. Only local buckling was evaluated, more precisely the utilization ratios of all tubes, where the output is the maximum utilization ratio in the jacket. Punching shear resistance of tubular joints is not considered in the surrogate model, because it is not part of the pre-design process. Steel with a yield stress of $355\,\mathrm{MPa}$ (S355) is considered

---

[3]For some extreme load events, this is a rather low value. However, due to limited capacity of computational resources, it was decided to choose this length for all simulations.





**Table 3.** Considered design load sets according to IEC-61400-3 (International Electrotechnical Commision, 2009) for the ultimate limit state (SF: partial safety factor, $v_s$: mean wind speed, TI: turbulence intensity, $H_s$: significant wave height, $T_p$: wave peak period, $\theta_{wind}$: wind direction, $\theta_{wave}$: wave direction, $u_w$: near-surface current velocity, $u_{ss}$: sub-surface current velocity, MSL: mean sea level). Yaw error is constantly set to $-8°$, if not stated differently.

| DLC | Quantity | Wind | Waves | Directionality | Current | Water Level | Special Event |
|---|---|---|---|---|---|---|---|
| 1.3 SF = 1.35 | 1 | $v_s = 15.40\,\mathrm{m\,s^{-1}}$ TI = 58.10% | $H_S = 2.04\,\mathrm{m}$ $T_p = 7.50\,\mathrm{s}$ | $\theta_{wind} = 0°$ $\theta_{wave} = 0°$ | $u_w(0) = 0.42\,\mathrm{m\,s^{-1}}$ $u_{ss}(0) = 0\,\mathrm{m\,s^{-1}}$ | MSL | |
| 1.3 SF = 1.35 | 1 | $v_s = 15.40\,\mathrm{m\,s^{-1}}$ TI = 58.10% | $H_S = 2.04\,\mathrm{m}$ $T_p = 7.50\,\mathrm{s}$ | $\theta_{wind} = 15°$ $\theta_{wave} = 15°$ | $u_w(0) = 0.42\,\mathrm{m\,s^{-1}}$ $u_{ss}(0) = 0\,\mathrm{m\,s^{-1}}$ | MSL | |
| 1.3 SF = 1.35 | 1 | $v_s = 17.40\,\mathrm{m\,s^{-1}}$ TI = 44.22% | $H_S = 2.50\,\mathrm{m}$ $T_p = 7.50\,\mathrm{s}$ | $\theta_{wind} = 0°$ $\theta_{wave} = 0°$ | $u_w(0) = 0.42\,\mathrm{m\,s^{-1}}$ $u_{ss}(0) = 0\,\mathrm{m\,s^{-1}}$ | MSL | |
| 1.6a SF = 1.35 | 1 | $v_s = 11.40\,\mathrm{m\,s^{-1}}$ TI = 8.09% | $H_S = 10.60\,\mathrm{m}$ $T_p = 15.09\,\mathrm{s}$ | $\theta_{wind} = 0°$ $\theta_{wave} = 0°$ | $u_w(0) = 0.42\,\mathrm{m\,s^{-1}}$ $u_{ss}(0) = 0\,\mathrm{m\,s^{-1}}$ | MSL +2.02 m | |
| 2.3 SF = 1.1 | 1 | $v_s = 25.00\,\mathrm{m\,s^{-1}}$ TI = 8.09% | $H_S = 4.63\,\mathrm{m}$ $T_p = 10.47\,\mathrm{s}$ | $\theta_{wind} = 0°$ $\theta_{wave} = 0°$ | $u_w(0) = 0.42\,\mathrm{m\,s^{-1}}$ $u_{ss}(0) = 0\,\mathrm{m\,s^{-1}}$ | MSL | Grid loss |
| 2.3 SF = 1.1 | 1 | $v_s = 25.00\,\mathrm{m\,s^{-1}}$ TI = 8.09% | $H_S = 4.63\,\mathrm{m}$ $T_p = 10.47\,\mathrm{s}$ | $\theta_{wind} = 60°$ $\theta_{wave} = 60°$ | $u_w(0) = 0.42\,\mathrm{m\,s^{-1}}$ $u_{ss}(0) = 0\,\mathrm{m\,s^{-1}}$ | MSL | Grid loss |
| 6.1a SF = 1.35 | 1 | $v_s = 42.14\,\mathrm{m\,s^{-1}}$ TI = 12.47% | $H_S = 4.63\,\mathrm{m}$ $T_p = 10.47\,\mathrm{s}$ | $\theta_{wind} = 0°$ $\theta_{wave} = 0°$ | $u_w(0) = 1.88\,\mathrm{m\,s^{-1}}$ $u_{ss}(0) = 0.69\,\mathrm{m\,s^{-1}}$ | MSL +2.74 m | |
| 6.2a SF = 1.1 | 1 | $v_s = 42.14\,\mathrm{m\,s^{-1}}$ TI = 12.47% | $H_S = 4.63\,\mathrm{m}$ $T_p = 10.47\,\mathrm{s}$ | $\theta_{wind} = 0°$ $\theta_{wave} = 0°$ | $u_w(0) = 1.88\,\mathrm{m\,s^{-1}}$ $u_{ss}(0) = 0.69\,\mathrm{m\,s^{-1}}$ | MSL +2.74 m | Yaw error 60° |
| 6.2a SF = 1.1 | 1 | $v_s = 42.14\,\mathrm{m\,s^{-1}}$ TI = 12.47% | $H_S = 4.63\,\mathrm{m}$ $T_p = 10.47\,\mathrm{s}$ | $\theta_{wind} = 0°$ $\theta_{wave} = 0°$ | $u_w(0) = 1.88\,\mathrm{m\,s^{-1}}$ $u_{ss}(0) = 0.69\,\mathrm{m\,s^{-1}}$ | MSL +2.74 m | Yaw error 90° |
| 6.2a SF = 1.1 | 1 | $v_s = 42.14\,\mathrm{m\,s^{-1}}$ TI = 12.47% | $H_S = 4.63\,\mathrm{m}$ $T_p = 10.47\,\mathrm{s}$ | $\theta_{wind} = 0°$ $\theta_{wave} = 0°$ | $u_w(0) = 1.88\,\mathrm{m\,s^{-1}}$ $u_{ss}(0) = 0.69\,\mathrm{m\,s^{-1}}$ | MSL +2.74 m | Yaw error 120° |





as the material for the entire structure, excluding structural appurtenances. The final output value of the structural code check is the maximum utilization ratio among all considered load cases, including partial safety factors.

## 4.5 Derivation and Parametrization of Gaussian Process Regression Models

While the outputs of both limit state assessments are single real values, it has to be conceived that the output values are distributed differently. GPR models are mainly governed by the kernel function choice and the corresponding hyperparameters. Different kernel functions were tested with respect to the creation of appropriate surrogate models and evaluated in terms of cross-validations in this section. Due to the highly nonlinear character of the utilized structural codes and therefore significant variance in the model outputs, a certain extent of uncertainty has to be tolerated. For the learning procedure, the fatigue damages are logarithmized, because the underlying S-N-curve is also logarithmic and the range of values covers at least four powers of ten. For the ultimate limit state, results cover only a range from zero to about three, no normalization is necessary. However, to exclude severe outliers from the training set of the surrogate model for the ultimate limit state, $10\%$ of the samples with the highest extreme load utilization ratios are excluded.

The problem of choosing the right kernel function is discussed by many authors. In order to limit the extent of this section, the reader is referred to the works of Duvenaud (2014) and King (2016) for further details. In general, the kernel choice implies a belief about the shape or smoothness of the covariance. In this case, four commonly used, stationary kernel functions are compared that represent relatively smooth approximations of the function.

The squared exponential kernel reads:

$$k_{SE}(\boldsymbol{x}, \boldsymbol{x}') = \exp\left(-\frac{(\boldsymbol{x} - \boldsymbol{x}')(\boldsymbol{x} - \boldsymbol{x}')^T}{2l^2}\right), \tag{38}$$

the Matérn 3/2 kernel:

$$k_{Ma_{3/2}}(\boldsymbol{x}, \boldsymbol{x}') = \left(1 + \frac{\sqrt{3(\boldsymbol{x} - \boldsymbol{x}')(\boldsymbol{x} - \boldsymbol{x}')^T}}{l}\right) \exp\left(-\frac{\sqrt{3(\boldsymbol{x} - \boldsymbol{x}')(\boldsymbol{x} - \boldsymbol{x}')^T}}{l}\right), \tag{39}$$

the Matérn 5/2 kernel:

$$k_{Ma_{5/2}}(\boldsymbol{x}, \boldsymbol{x}') = \left(1 + \frac{\sqrt{5(\boldsymbol{x} - \boldsymbol{x}')(\boldsymbol{x} - \boldsymbol{x}')^T}}{l} + \frac{5(\boldsymbol{x} - \boldsymbol{x}')(\boldsymbol{x} - \boldsymbol{x}')^T}{3l^2}\right) \exp\left(-\frac{\sqrt{5(\boldsymbol{x} - \boldsymbol{x}')(\boldsymbol{x} - \boldsymbol{x}')^T}}{l}\right), \tag{40}$$

and the rational quadratic kernel:

$$k_{RQ}(\boldsymbol{x}, \boldsymbol{x}') = \left(1 + \frac{(\boldsymbol{x} - \boldsymbol{x}')(\boldsymbol{x} - \boldsymbol{x}')^T}{2al^2}\right)^a, \tag{41}$$

where $l$ is a length scale and $a$ a weighting parameter. It is best practice to choose different scales for all input parameters. This is called automatic relevance determination (Duvenaud, 2014).

The squared exponential kernel is a common choice for Gaussian processes as an "initial guess,"because it is infinitely differentiable and therefore very smooth. The Matérn kernels are less smooth than the squared exponential kernel, where Matérn 3/2 is once and Matérn 5/2 twice differentiable. The rational quadratic kernel is a sum of squared exponential kernels





**Table 4.** Cross-validation results for kernel functions applied to fatigue and ultimate limit state outputs, each case with ideal hyperparameters obtained by maximum likelihood estimation.

| Limit State | Cross-Validation Type | Error Type | Kernel Function | | | |
| --- | --- | --- | --- | --- | --- | --- |
| | | | $k_{SE}$ | $k_{Ma_{3/2}}$ | $k_{Ma_{5/2}}$ | $k_{RQ}$ |
| Fatigue | Leave-one-out | $e_{bias}$ | $-0.003$ | $-0.002$ | $-0.003$ | $-0.002$ |
| | | $e_{mse}$ | $0.052$ | $0.044$ | $0.043$ | $0.041$ |
| | Ten-fold | $e_{bias}$ | $-0.007$ | $-0.004$ | $-0.003$ | $-0.005$ |
| | | $e_{mse}$ | $0.073$ | $0.049$ | $0.049$ | $0.047$ |
| | Five-fold | $e_{bias}$ | $0.004$ | $-0.008$ | $0.000$ | $-0.012$ |
| | | $e_{mse}$ | $0.084$ | $0.062$ | $0.063$ | $0.061$ |
| Ultimate | Leave-one-out | $e_{bias}$ | $0.003$ | $0.001$ | $0.003$ | $-0.002$ |
| | | $e_{mse}$ | $0.057$ | $0.053$ | $0.054$ | $0.057$ |
| | Ten-fold | $e_{bias}$ | $0.004$ | $0.006$ | $0.006$ | $-0.002$ |
| | | $e_{mse}$ | $0.056$ | $0.055$ | $0.056$ | $0.061$ |
| | Five-fold | $e_{bias}$ | $0.003$ | $0.002$ | $0.003$ | $0.008$ |
| | | $e_{mse}$ | $0.055$ | $0.056$ | $0.054$ | $0.061$ |

with the capability to weight between large- and small-scale variations. To figure out which kernel function is most suitable for both surrogate models, various cross-validations are performed. A $N$-fold cross-validation means that the training data set (which comprises 200 jacket samples in the fatigue limit state case and 180 samples in the ultimate limit state case in this study) is divided into $N$ parts with equal size. $N-1$ parts are then used to train the model and the leftover is the test set, which

5    is used to predict a vector of validation results, $\boldsymbol{y}^*$. This is repeated $N$ times to compute the mean of the two common error measures bias $e_{bias}$ and mean square error $e_{mse}$:

$$e_{bias} = \frac{1}{N} \sum_{n=1}^{N} \left( \frac{1}{M} \sum_{m=1}^{M} \left( \boldsymbol{y}^*_{n,m} - \boldsymbol{y}_{n,m} \right) \right), \tag{42}$$

$$e_{mse} = \frac{1}{N} \sum_{n=1}^{N} \left( \frac{1}{M} \sum_{m=1}^{M} \left( \boldsymbol{y}^*_{n,m} - \boldsymbol{y}_{n,m} \right)^2 \right), \tag{43}$$

where $\boldsymbol{y}^*_{n,m}$ is the the $m$th predicted element in the $n$th cross-validation set and $\boldsymbol{y}_{n,m}$ is the corresponding value in the output

10    vector. $M$ is the size of the cross-validation leftover. For instance, in the case of a ten-fold cross-validation, $M$ is 20. While the mean square error is always positive, the bias can have both positive and negative values. Table 4 shows validation results for the four kernels using leave-one-out, ten-fold, and five-fold cross-validations. There are no values completely off and all kernel functions lead to similar results in the fatigue limit state case, where the Matérn 5/2 function is eventually chosen for both surrogate models.



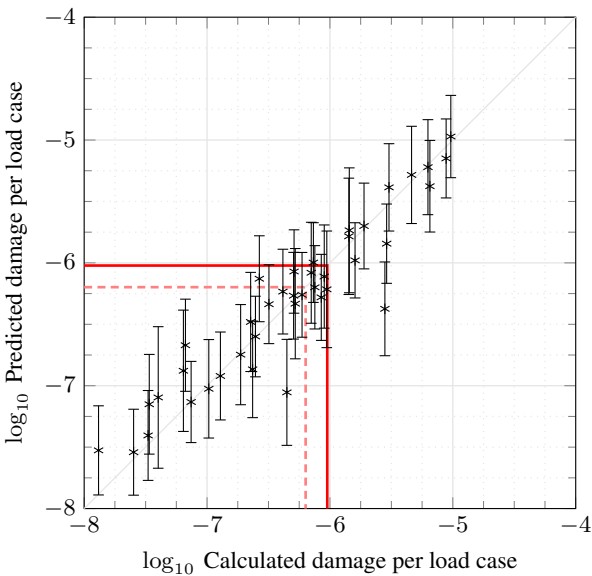 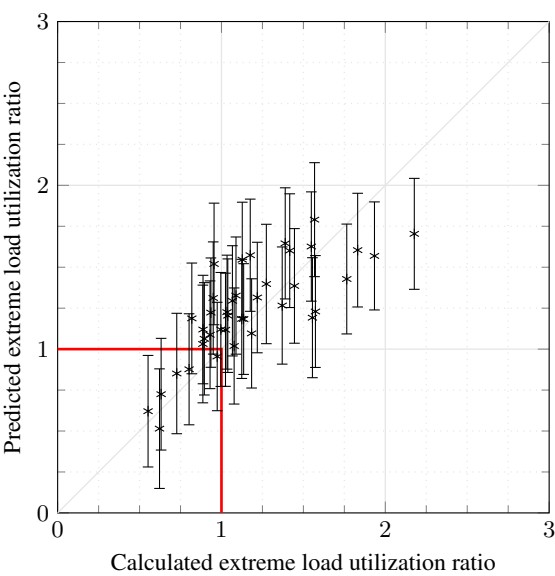

**Figure 6.** Prediction results for all samples of the validation set. Asterisks depict mean predicted damages in the first and mean extreme load utilization ratios in the second plot, whisker ranges illustrate the 95% significance intervals. —— illustrates the critical damage related to a 20-year lifetime of the structure or a utilization ratio of 1, respectively. Moreover, the 30-year damage is illustrated with - - - in the first plot.

### 4.6 Validation of Gaussian Process Regression Models

Based on the kernel function selection, the surrogate models are validated with a new Latin Hypercube sample from the design space given in Table 1, which comprises 40 jacket designs. A Matérn 5/2 kernel with independent hyperparameters and a Gaussian likelihood function with $\ln\left(\sqrt{\sigma_n^2}\right) = -2.06$, where $\sqrt{\sigma_n^2}$ is the mean standard deviation of logarithmized damage
5   per load case accounting for load set reduction uncertainty, evaluated from the results by Häfele et al. (2017a, b), are chosen for the fatigue case. The ultimate limit state case does not incorporate prior knowledge of uncertainty, because it is assumed that one of the considered load cases in Table 3 is the severest imaginable one. The predicted validation values, for both fatigue and ultimate limit state, are shown in Figure 6. Although the drawn whiskers show quite wide prediction intervals, the mean values predict the calculated ones well in both diagrams. Therefore, it can be stated that Gaussian process regression is suitable
10   for this task.

### 5   Example

Although the focus of this work shall not be a comprehensive design study, a short example is provided in this section, which shows how the proposed models can be used in further studies.





**Table 5.** Reference unit costs for the regarded example, a 5-MW reference turbine in a water depth of $50\,\mathrm{m}$.

| Unit Cost | Unit | Mean | Standard Deviation |
|-----------|------|------|--------------------|
| $a_1$ | $\mathrm{kg}^{-1}$ | $1.0$ | $5.0 \times 10^{-2}$ |
| $a_2$ | $\mathrm{m}^{-3}$ | $4.0 \times 10^6$ | $0.5 \times 10^6$ |
| $a_3$ | $\mathrm{m}^{-2}$ | $1.0 \times 10^2$ | $1.0 \times 10^1$ |
| $a_4$ | $\mathrm{m}^{-1}$ | $2.0 \times 10^4$ | $5.0 \times 10^3$ |
| $a_5$ | $\mathrm{kg}^{-1}$ | $1.0$ | $2.5 \times 10^{-1}$ |
| $a_6$ | $-$ | $2.0 \times 10^5$ | $5.0 \times 10^4$ |
| $a_7$ | $-$ | $1.0 \times 10^5$ | $2.5 \times 10^4$ |

We assume that for a fixed wind farm location with 50-m water depth, NREL 5-MW turbine, FINO3 environmental conditions, and OC3 soil properties, it has to be evaluated which of three given jacket designs is most suitable with regard to capital expenses. There is uncertainty in the capital expenditures arising from the market situation, the availability of fabrication facilities and ships, the distance of the installation site from shore, the weather situation and sea state, etc. It is assumed

that this uncertainty can be described in terms of normally distributed cost model parameters, given as mean values[4] and standard deviations in Table 5. The parameter distributions indicate relatively high uncertainty, in particular in the expenses for transport and installation, which is a common experience in the wind farm planning process. There are three substructure options to be compared: the first (1), derived from the so-called OC4-jacket Popko et al. (2014), and second (2) ones are four-legged ($N_X = 4$) jackets, the third (3) one is a three-legged ($N_X = 3$) structure. All structures have a length of $L = 70\,\mathrm{m}$

with transition piece $L_{MSL} = 20\,\mathrm{m}$ above mean sea level and use steel ($E = 2.100 \times 10^{11}\,\mathrm{N\,m^{-2}}$, $G = 8.077 \times 10^{10}\,\mathrm{N\,m^{-2}}$, $\rho = 7.850 \times 10^3\,\mathrm{kg\,m^{-3}}$) as the material. The height between the ground and lowermost K-joint layer is $L_{OSG} = 5\,\mathrm{m}$, and the transition piece height is $L_{TP} = 4\,\mathrm{m}$. Furthermore, all jackets have mud braces ($x_{MB} = $ true), the foot radii, $R_{foot}$, are all $8.485\,\mathrm{m}$, the bay height ratio, $q$, is $0.8$, and the head-to-foot radius ratio, $\xi$, is $0.67$. The leg radius-to-thickness and the leg-to-brace thickness ratios are held constant at $\gamma = \gamma_b = \gamma_t = 15.0$ and $\tau = \tau_b = \tau_t = 0.5$, respectively. The structures differ—

except for the number of legs ($N_L$)—in the number of bays ($N_X$) and tube dimensions ($D_L, \beta_b, \beta_t$). The first one has four bays, a leg diameter of $1.2\,\mathrm{m}$, $\beta = \beta_b = \beta_t = 0.67$. The second one has only three bays, but higher tube diameters and thicknesses with $D_L = 1.32\,\mathrm{m}$ and constant $\beta = \beta_b = \beta_t = 0.75$. The third jacket is the same as the first one, but with only three legs ($N_L = 3$) and an increased leg diameter $D_L = 1.44\,\mathrm{m}$. Thus, all structures are representative for different approaches known from practical applications and it is easily imaginable that they differ in all cost factors of the cost model except for the fixed

expenses.

Firstly, the cost contributions $C_1 \ldots C_7$ are calculated for each substructure according to the proposed cost model. Now, two helpful properties are used to evaluate the costs:

---

[4]The values are in accordance with practical experience and published information about jacket expenditures (Michels, 2014; National Wind Technology Center Information Portal, 2014).

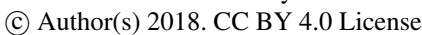



**Figure 7.** Probability density functions (PDFs) of the cost contributions $C_1 \dots C_7$, the total expenses $C_{total}$, the (logarithmized) predicted damage per load case, and the predicted extreme load utilization ratio for the regarded example case. Each plot shows the probability density in dependence of cost, damage exponent, and utilization ratio, respectively. Structure (1) is illustrated by ——, structure (2) by – – –, and structure (3) by ⋯⋯. —— illustrates the critical damage related to a 20-year lifetime of the structure or a utilization ratio of 1, respectively. Moreover, the 30-year damage is illustrated with – – –.



1. The total costs of each substructure, $C_{total}$, are a linear combination of the single contributions, $C_1 \dots C_7$,

2. The sum of normal distributions is again normally distributed.

Figure 7 shows the resulting probability density functions of all cost contributions and the total expenses, when the normally distributed unit costs in Table 5 are combined with the proposed cost model. The three-leg design (3) is the cheapest among the considered structures, because the tube dimension's increase of $20\%$ (all tube sizing parameters depend linearly on the leg diameter) is overcompensated by the the reduction in jacket legs, which shows in the factor mass, resulting in significantly lower costs. The structures (1) and (2) show stronger similarities in all cost contributions, adding up in nearly equal total expenses.

However, the cost assessment is not meaningful without consideration of structural code checks. The surrogate models for fatigue and ultimate limit state are utilized for prediction and also shown in Figure 7. Structure (1) takes a mean damage of $10^{-6.72}$ per load case, structure (2) $10^{-6.91}$, and structure (3) $10^{-6.72}$, all with similar variance. Linear damage accumulation (implying that the lifetime is reached at a cumulated damage of $1$) and a simulation time per load case of $10\,\mathrm{min}$ yields lifetimes of approximately $100\,\mathrm{years}$, $155\,\mathrm{years}$, and $100\,\mathrm{years}$ for the three structures, considering a fatigue safety factor of $1.25$. The same procedure is applied to ultimate limit state assessment and mean tube utilization ratios of $1.05$, $0.72$, and $0.94$ are obtained in the critical load case for the structures (1), (2), and (3), respectively. Therefore, although all structures are quite close to an ideal utilization ratio, the second structure has the highest capacity concerning extreme loads.

Although only three designs were considered in this example, it is conceivable that three-legged structures are truly competitive with respect to the given boundaries, because the design of structure (3) is related to the lowest capital expenses and has sufficient load capacities in the fatigue and ultimate limit state. According to the proposed cost model, the cost saving arises mainly from two contributions, namely transition piece expenses and foundation and installation expenses, both depending linearly on the number of jacket legs. This is in agreement with experiences from practical applications, because three-legged structures have recently increased in importance, visible in the number of offshore installations for turbines with intermediate rated power. Comparing structure (1) and (2), the cost differences are marginal, while structure (2) turns out to be much better in terms of structural properties, visible in a higher lifetime and a lower extreme load utilization ratio. Therefore, it can be stated that the number of jacket legs and the leg diameter (in the case of dependent brace dimensions) are key parameters in the first phase of jacket design. A quantitative sensitivity analysis of the remaining parameters has to be conducted in forthcoming studies.

It can be imagined that the approach is easily usable for far more complex studies, where the number of design samples is much higher than in the present example, because the entire procedure—which usually requires enormous numerical capacity—was solved in a negligible amount of time. It was already discussed that every jacket design requires a high number of time domain simulations to perform structural code checks. Therefore, the proposed methodology is appropriate to assess the topology and dimensions of a substructure, while structural details still have to be determined with high-fidelity models.

Moreover, the example shows that uncertainty can be easily incorporated in the design assessment using the proposed models for capital expenses and structural code checks. This may lead to probabilistic studies or robust jacket design.



## 6   Limitations

The models established in this work provide groundwork to regard the jacket design process from a scientific point of view, not from an application-oriented design perspective, which depends highly on (human) expert knowledge. This aspect is emphasized strongly at this point, because the outcome from studies based on these models will most likely not represent the
geometry of the final structure, but an initial or conceptual design approach, suitable for implementations with high numerical demands. Therefore, although the proposed models provide a comprehensive basis for design evaluations or optimization, they have to be used with caution. There is still a distinct amount of uncertainty in the surrogate model outputs, which arises from different sources, such as load set reductions, relatively small training sets (due to limited numerical capacity), or nonlinearities in physical models or structural code checks.

In addition, it has to be mentioned that, though the methods are probably applicable to other turbines as well, the numerical parameters and results in the regarded example are only valid for a jacket substructure at a given (fictive) offshore location with a $50$-$\mathrm{m}$ water depth, FINO3 environmental conditions, and the NREL 5-MW turbine. An adaption to different boundaries requires a reestimation of the parameters.

## 7   Conclusions

The objective of this work was to provide a minimal, but comprehensive approach to conceptual studies on jacket substructures for wind turbines. For this purpose, a geometry model was defined. A completely analytical cost model was derived afterwards. The issue of computationally expensive structural code checks was faced by surrogate modeling, namely Gaussian process regression models. Finally, an example was regarded to show the capabilities of the developed models, where three artificial structures were analysed. It was shown that different jacket design approaches (varying in topology and tube dimensions) may
be appropriate solutions for a given wind turbine and environmental conditions.

Deliberately, this paper does not provide too extensive numerical results for applied science. The proposed models and equations are to be used for more realistic design studies on latticed substructures for offshore wind farms. Therefore, the path can continue in two ways: first, design studies, not focused on the structural aspects can benefit from these models, because they do not require too much knowledge about physical details. But second and intentionally, this work contributes a substantial
improvement to jacket optimization approaches, yet mostly focuses on tube dimensioning and often neglects structural topology aspects, a correct cost assessment, or realistic structural code checks. In particular, the utilization of surrogate modeling is very promising, when dealing with meta-heuristic algorithms like evolutionary or swarm-based approaches applied to the jacket optimization problem, because the related numerical expenses are significantly lower compared to approaches based on time domain simulations. This may lead to much more detailed analyses of the optimization procedure from the mathematical point
of view, because approaches known from literature are focusing on technical aspects. Questions to be answered in this context are, for instance, how the constraints can be handled efficiently or which algorithm is most suitable for the jacket optimization problem.





*Data availability.* The results of training and validation sets, including fatigue and ultimate limit state code checks, are provided and available for research purposes. To set up Gaussian process regression models, we use and recommend using GPML, which can be run with Octave, MATLAB, or Python. GPML is freely available at http://www.gaussianprocess.org/gpml/code/matlab/doc/.

*Competing interests.* No competing interests are present in this study.

*Acknowledgements.* This work was supported by the compute cluster, which is funded by the Leibniz Universität Hannover, the Lower Saxony Ministry of Science and Culture (MWK), and the German Research Foundation (DFG).

Cordial thanks are given to Jason Jonkman, Amy Robertson, and Katherine Dykes from the National Renewable Energy Laboratory, who supported this work with many valuable remarks and suggestions, and Manuela Böhm from Leibniz Universität Hannover for supporting the numerical simulation work.

The Alliance for Sustainable Energy, LLC (Alliance) is the manager and operator of the National Renewable Energy Laboratory (NREL). NREL is a national laboratory of the U.S. Department of Energy, Office of Energy Efficiency and Renewable Energy. This work was authored by the Alliance and supported by the U. S. Department of Energy under Contract No. DE-AC36-08GO28308. Funding was provided by the U.S. Department of Energy Office of Energy Efficiency and Renewable Energy, Wind Energy Technologies Office. The views expressed in the article do not necessarily represent the views of the U.S. Department of Energy or the U.S. government. The U.S. government retains, and

the publisher, by accepting the article for publication, acknowledges that the U.S. government retains a nonexclusive, paid-up, irrevocable, worldwide license to publish or reproduce the published form of this work, or allow others to do so, for U.S. government purposes.





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
