# Peer review of "A Systematic Approach to Offshore Wind Turbine Jacket Pre-Design and Optimization: Geometry, Cost, and Surrogate Structural Code Check Models"

_Wind Energy Science, 2018_

## Referee Comment (RC1) · Anonymous Referee #1 · 23 May 2018

This paper describes a systematic strategy to provide a basic offshore wind turbine jacket model that can be efficiently applied in conceptual studies and optimization techniques. This strategy presents a basis for more realistic designs and mainly using mathematically manageable equations. The result of this work is suitable for scientific and industrial applications. Also, the contents of this manuscript are interesting. However, I think the author should also address the following issues for improving the quality of this paper: (1) Could you describe your academic contributions for this approach? Or could you explain the difference between your theory and the others to

reflect the "new". (Briefly) (2) In abstract, (The outcome can be utilized for preliminary design studies "and" jacket optimization schemes "and" is suitable for scientific "and" industrial applications.)  needs to be revised. (3) Mathematical equations are derived by what criteria? There is not any reference. For example: equations [10-12], [19-20] and [24] . . ..  (4) In the text and content of the figures, use Latin letters to represent concepts or sub-sections.  On page 25, line 1 and 2; (1, 2) :(a, b) Figure 2: (1-2-3-4):(a,b,c.d) (5) On page 18, line 13 What is the "DNV GL RP-0005 (?)" ? (6) On page 20, line 27 Wrong Correct "initial guess," "initial guess",

---

## Referee Comment (RC2) · Anonymous Referee #1 · 29 May 2018

I am happy to see that all the comments and questions have been addressed in the revision. I would like to suggest the acceptance and publication.

---

## Author Comment (AC1) · 29 May 2018

We thank the referee for his/her valuable comments.

Find below the list of changes and some comments. The revised document is attached to this comment.

(1) This is probably mistakable. The paragraph from page 4, line 24 to page 5, line 10 summarizes five common problems, from our point of view, leading to nonoptimal structures. As stated on page 5, line 15ff. the present approach addresses these

issues: "The goal of the current work is to provide a basic jacket model that can be efficiently used in conceptual studies and optimization approaches by preventing the issues stated above and providing a basis for more realistic designs and mainly using mathematically manageable equations." To highlight this aspect, it is repeated in the conclusion section in the revised version. We hope that this point is resolved in this way. (2) This is corrected in the revised version. (3) Most equations have no reference. However, in the revised version the explanations of some equations are more detailed: - Equations (10), (11), and (12) can be derived from Figure 3. This important reference was missing and has been added. - The explanation of equations (19), (20), and (21) has been improved. - The meaning of each addend in the equations (24), (25), and (26) and some explanations have been added. (4) We use latin letters in Figure 2 (a-b-c-d) and the example (a-b-c) now. (5) This is corrected in the revised version. (6) This is corrected in the revised version.

Please also note the supplement to this comment:
https://www.wind-energ-sci-discuss.net/wes-2018-39/wes-2018-39-AC1-supplement.pdf

**Supplement:**

[revised manuscript text omitted]

---

## Referee Comment (RC3) · Anonymous Referee #2 · 16 Jul 2018

This paper presents an approach to offshore wind turbine jacket pre-design and optimisation. A jacket model and a cost model are developed in this work.

Specific comments are as follows.

1. Introduction

The novelty/contribution of the paper is unclear and should be highlighted in the introductory section.

4.1. Training and Validation Data Sets

Please justify the values used for jacket model parameter boundaries in Table 1.

**4.3. Time Domain Simulations**

Please elaborate how the soil is modelled in the model to take account of the soil-structure interaction.

**4.4. Post-Processing of Time Domain Results**

Please justify the choice of S-N curve used in the study.

Please provide the references for values of wind and current speeds as well as the values for significant wave height in Table 3.

**5. Example**

Please justify why normal distribution is used for all cost model parameters.

Please justify the values of standard deviation in Table 5.

---

## Author Comment (AC2) · 20 Jul 2018

We appreciate the second reviewer's comments that are addressed in the following.

- *"The novelty/contribution of the paper is unclear and should be highlighted in the introductory section."*
  The introduction section is reformulated in the revised version, i.e., the main innovation is explicitly stated now. This was already mentioned in the comments of the first reviewer. Together with the response in author comment AC2, we believe

that this resolves the issue.

- *"Please justify the values used for jacket model parameter boundaries in Table 1."*
Basis for the choice of these parameters was the OC4-jacket. This very important reference was indeed missing and has been to the revised version.

- *"Please elaborate how the soil is modelled in the model to take account of the soil-structure interaction."*
A brief description, how the soil-structure interaction approach works, has been added to the corresponding section.

- *"Please justify the choice of S-N curve used in the study."*
Fatigue checks are only performed for tubular joints, S-N-curve DNV-T. The paragraph was slightly reformulated to resolve this issue.

- *"Please provide the references for values of wind and current speeds as well as the values for significant wave height in Table 3"*
The reference was missing and has been added.

- *"Please justify why normal distribution is used for all cost model parameters."*
This is a scientific example. Therefore, normal distributions are used to consider uncertainty to a certain extent. We cannot prove that the assumption of normally distributed cost values is correct, but we believe that the example is easily transferable to any other distribution of unit costs.

- *"Please justify the values of standard deviation in Table 5."*
This is similar to the previous point. The given values are determined by expert knowledge and there is no reference. Based on expert knowledge, we tried to define an example where the scatter of cost units is in a realistic dimension, which shows in the values of standard deviation (for instance, material price is

supposed to be constant, while transport and installation costs vary strongly). To resolve this and the previous point, we have modified the example section slightly to clarify this point.

We have attached a revised document that includes all changes to this response and hope that all issues are resolved in this way.

Please also note the supplement to this comment:
https://www.wind-energ-sci-discuss.net/wes-2018-39/wes-2018-39-AC2-supplement.pdf

**Supplement:**

[revised manuscript text omitted]

---

## Author Response (AR1)

**"A Systematic Approach to Offshore Wind Turbine Jacket Pre-Design and Optimization: Geometry, Cost, and Surrogate Structural Code Check Models"**

by Jan Häfele et al.

**Point-by-point response to the reviewers including list of changes**

**Comments of reviewer #1 (RC1) and responses**

*(1) "Could you describe your academic contributions for this approach? Or could you explain the difference between your theory and the others to reflect the "new". (Briefly)"*

This is probably mistakable. The paragraph from page 4, line 24 to page 5, line 10 summarizes five common problems, from our point of view, leading to nonoptimal structures. As stated on page 5, line 15ff. The present approach addresses these issues: "The goal of the current work is to provide a basic jacket model that can be efficiently used in conceptual studies and optimization approaches by preventing the issues stated above and providing a basis for more realistic designs and mainly using mathematically manageable equations." To highlight this aspect, it is repeated in the conclusion section in the revised version.  We hope that this point is resolved in this way.

*(1) In abstract, (The outcome can be utilized for preliminary design studies "and" jacket optimization schemes "and" is suitable for scientific "and" industrial applications.) needs to be revised.*

This is corrected in the revised version.

(2) *"Mathematical equations are derived by what criteria? There is not any reference. For example: equations [10-12], [19-20] and [24]…"*

Most equations have no reference. However, in the revised version the explanations of some equations are more detailed:

- Equations (10), (11), and (12) can be derived from Figure 3. This important reference was missing and has been added.
- The explanation of equations (19), (20), and (21) has been improved.
- The meaning of each addend in the equations (24), (25), and (26) and some explanations have been added.

*(3) "In the text and content of the figures, use Latin letters to represent concepts or sub-sections.  On page 25, line 1 and 2;  (1, 2) :(a, b) Figure 2:  (1-2-3-4):(a,b,c.d)"*

We use latin letters in Figure 2 (a-b-c-d) and the example (a-b-c) now.

*(4) "On page 18, line 13 What is the "DNV GL RP-0005 (?)" ?"*

This is corrected in the revised version.

*(5) "On page 20, line 27 Wrong Correct "initial guess," "initial guess","*

This is corrected in the revised version.

**Comments of reviewer #2 (RC3) and responses**

*(1) "The novelty/contribution of the paper is unclear and should be highlighted in the introductory section."*

The introduction section is reformulated in the revised version, i.e., the main innovation is explicitly stated now. This was already mentioned in the comments of the first reviewer. Together with the response in author comment AC2, we believe that this resolves the issue.

*(2) "Please justify the values used for jacket model parameter boundaries in Table 1."*
Basis for the choice of these parameters was the OC4-jacket. This very important reference was indeed missing and has been to the revised version.

*(3) "Please elaborate how the soil is modelled in the model to take account of the soil-structure interaction."*

A brief description, how the soil-structure interaction approach works, has been added to the corresponding section

*(4) "Please justify the choice of S-N curve used in the study."*

Fatigue checks are only performed for tubular joints, S-N-curve DNV-T. The paragraph was slightly reformulated to resolve this issue.

*(5) "Please provide the references for values of wind and current speeds as well as the values for significant wave height in Table 3."*

The reference was missing and has been added.

*(6) "Please justify why normal distribution is used for all cost model parameters."*

This is a scientific example. Therefore, normal distributions are used to consider uncertainty to a certain extent. We cannot prove that the assumption of normally distributed cost values is correct, but we believe that the example is easily transferable to any other distribution of unit costs.

*(7) "Please justify the values of standard deviation in Table 5.""*

This is similar to the previous point. The given values are determined by expert knowledge and there is no reference. Based on expert knowledge, we tried to define an example where the scatter of cost units is in a realistic dimension, which shows in the values of standard deviation (for instance, material price is supposed to be constant, while transport and installation costs vary strongly). To resolve this and the previous point, we have modified the example section slightly to clarify this point.

**List of changes**

- Abstract was slightly modified (one sentence split).
- Section 1: Introduction was modified to emphasize the main innovation of the work.
- Section 4: Added information about the transition piece.
- Subsection 4.1: Added information, where parameter boundaries come from and reference to OC4-jacket report.
- Subsection 4.2: Added reference to data basis for extreme loads.
- Subsection 4.3: Added information about soil-structure interaction approach.
- Subsection 4.4: Added fatigue assessment details; reformulated second paragraph.
- Section 5: Reformulated to state clearly that normal distributions of cost parameters are an assumption and not based on real data.
- Section 7: Conclusions section was improved.
- Equations or Explanations of equations 10-12, 19-20, 24-26 were added or improved.
- Latin letters are used to represent subfigures and concepts.
- Fixed issue with reference on page 18, line 13 (initial version).
- Fixed punctuation issue on page 20, line 27 (initial version).

[revised manuscript text omitted]